# Assessment of Radioactivity Level in the Terrestrial and Marine Organisms in Yangjiang and Its Adjacent Areas (China)

**DOI:** 10.3390/ijerph18168767

**Published:** 2021-08-19

**Authors:** Dongmei Li, Zhongchen Jiang, Li Zhao, Feng Zhao, Peng Zhou

**Affiliations:** 1South China Sea Environment Monitoring Center, State Oceanic Administration (SOA), Guangzhou 510300, China; lidmay@foxmail.com (D.L.); jcc1967@163.com (Z.J.); ronda1522@163.com (L.Z.); quietpeak@163.com (F.Z.); 2South China Sea Testing and Appraisal Center, State Oceanic Administration (SOA), Guangzhou 510300, China; 3Nansha Islands Coral Reef Ecosystem National Observation and Research Station, Guangzhou 510300, China; 4Key Laboratory of Marine Enironmental Survey Technology and Application, Ministry of Natural Resources of People’s Republic of China (MNR), Guangzhou 510300, China

**Keywords:** radioactivity, risk assessment, terrestrial and marine organisms, Yangjiang and the adjacent areas

## Abstract

In order to assess the radioactive level in the terrestrial and marine organisms in Yangjiang and the adjacent areas, ^90^Sr, gross beta and gamma-emitting radionuclides (^238^U, ^226^Ra, ^228^Th, ^226^Ra, ^40^K, ^137^Cs, ^51^Cr, ^55^Fe, ^54^Mn, ^58^Co, ^60^Co and ^65^Zn) were analyzed from 2011 to 2012. The annual effective doses were estimated in the high natural radioactive background areas in Yangjiang (HBRAYJ). The specific activities of ^238^U, ^228^Th, ^226^Ra, ^40^K and ^137^Cs in all organisms were <0.05–5.20, 0.30–14.50, 0.11–3.58, 11.1–148.0 and <0.003–0.088 Bq/kg, whilst ^51^Cr, ^55^Fe, ^54^Mn, ^58^Co, ^60^Co, ^65^Zn and ^110m^Ag were below the minimum detectable activity. ^90^Sr and gross beta specific activities were 20.0–143.0 and 0.021–0.316 Bq/kg. Results show that ^228^Th was significantly higher than ^238^U and ^226^Ra of natural U series in organisms due to the rich-Th soils in the HBRAYJ; ^228^Th, ^226^Ra, ^40^K, ^137^Cs and ^90^Sr were greatly lower than the limited concentrations in Chinese foods. The internal dose mainly contributes to natural ^40^K, ^226^Ra and ^228^Th. It is useful to provide some basic data and assess the radiological risk from the HBRAYJ and Yangjiang nuclear power plants in future.

## 1. Introduction

Yangjiang is one of the famous areas with high natural radioactive background areas (HBRAYJ) in China, where the radiation level is about three times more than that in its adjacent normal background radiation areas (CA). Two HBRA regions with ~540 km^2^ are Dong-anling region (including three villages of Yong’an, Huan’an and Liang’an) and Tongyou region [1,2,3,4]. Yangjiang nuclear power plants (i.e., YJNPP) base is situated at the eastern coast in Dongping in Yangjiang, where six nuclear power units (6 × 1080 MW) have been built at present. The YJNPP site is more than 30 km away from the HBRAYJ, so that the natural radioactivity in the YJNPP base is generally at the normal background level [5]. The No.1 nuclear power unit began to be built on 26 September 2007 and came into operation on 26 March 2014. At present, six nuclear power units had been put into commercial operation until 24 July 2019. The low-level anthropogenic radioactive liquid and gas from the YJNPP normal operation maybe influence directly and/or indirectly the surrounding terrestrial and marine environment. Some radionuclides (e.g., ^3^H in water and gamma-emitting radionuclides in marine sediments) and radiation effect have been studied near YJNPP [5,6,7].

It is known that nuclear power is a clean and high-quality energy that can improve China’s energy supply structure, ensure energy and economic security and protect the environment. The development of Yangjiang city was always restricted by power and infrastructure at past. The YJNPP is a nuclear power plant built within/near the HBRAYJ. Such examples of building nuclear power plants in high-level areas are rare in the world. The effects of low dose radiation on human in the HBRAYJ has been still an unsolved but very important issue. In addition, the effects of low dose radiation on non-human species has become a concern since the beginning of this century. Therefore, it is a necessary task to investigate systematically the current radioactive levels and assess scientifically the potential risk.

It is important to determine the radionuclides in the environment because they constitute a source of exposure of flora, fauna and humans. Assessments of such radiological consequences to the environment generally include consideration of the dose from ingestion of contaminated foodstuffs [8]. Here, some organisms were collected in Yangjiang and the adjacent areas during August 2011–April 2012, according to the local production and diet habits. This radioactive investigation in various species organisms is conducted firstly and systematically. Some items of ^238^U, ^228^Th, ^226^Ra, ^40^K, ^90^Sr, ^137^Cs, ^134^Cs, ^51^Cr, ^55^Fe, ^54^Mn, ^58^Co, ^60^Co, ^65^Zn, ^110m^Ag and gross beta were analyzed to estimate annual effective dose near the HBRAYJ and YJNPP. Just sampling time is after the Fukushima Nuclear Power Plants Accident (FNA) in March 2011, and before the YJNPP commercial operation. Therefore, our results are expected to provide some basic background data. The dose calculations should help the public quantitatively assessing the risk associated with consumption of these popular organisms. It is useful to the routine and emergency radioactive monitoring around the YJNNP. This study is also expected and help to assess the radiological risk from the HBRAYJ and Yangjiang nuclear power plants in future.

## 2. Materials and Methods

### 2.1. Sampling and Preparation

Samples were collected from the local field-cultivated areas or markets in Yangjiang and the adjacent areas (viz, Dongping, Dagou, Huangjiangwei, Zhapo, Xiachuan island) from 5 August 2011 to 3 December 2012 (Figure 1). These organisms included twitch grass, bok-choy, sweet potato, unhulled rice, turnip, sugarcane, chicken, freshwater/sea fish, sea shrimp and mollusks (eg. oysters, mussels), etc. All organisms are divided into six sorts based on the local diet, viz, Crops, Potatoes, Vegetables/fruits, Fodder, Meat/fish/shrimp. The measurement and date possess were performed in the Radioactivity Monitoring Laboratory of South China Sea Environment Monitoring Center, State Oceanic Administration (SOA), China.

### 2.2. Analytical Methods

For plant samples, 5.0~10 kg of samples were purchased from the peasant, transported to the laboratory under the 4 °C conditions. Twitch grass, bok-choy, sweet potato, turnip and sugarcane were washed, minced, aired and then weighted. The unhulled rice was aired and then weighted. Subsequently, these wet samples were dried at 105 ± 5 °C. Finally, the dried samples were put into a muffler furnace for carbonization at ≤250 °C for ~48 h, followed by ashing to white or greyish white at ≤450 °C. The ignition must avoid open flame or some radionuclides will escape [9]. The ash was milled, sifted using an 80 mesh, weighed, wrapped using PE bags to counting. For zoological samples, 10~20 kg of samples were collected or purchased from the peasant/fisherman, transported to the laboratory under the −20 °C conditions. Chicken (without head and internal organs), fish (with head, scales and internal organs), shrimp (with head, shell and meat) and mollusks without the hard shells (with head, gonads and internal organs) were washed, aired and then weighted. As described above, the samples were dried, ignited to ash, milled, sifted, weighed and then wrapped subsequently.

#### 2.2.1. Gamma-Emitting Radionuclides

The measurement was performed using a HPGe γ-spectrometer with Model 747 Lead Shield and Model BE5030 detector with its crystal diameter of 80.5 mm and thickness 31.0 mm (from Canberra Industries, Inc., Oak Ridge, TN, USA). Resolution (FWHM) at 5.9 keV (^55^Fe), 122 keV (^57^Co) and 1332 keV (^60^Co), are 0.500, 0.750 and 2.200 keV, respectively [10,11,12]. The multi-channel analyzer is Accuspec interface board coupling with microcomputer and conversion gain and memory are 8192. Genie-2000 software was used to analyze spectrum data.

Ash sample was packed tightly into plastic cylindrical containers (50/75 mm in diameter/height), weighed, measured for height, calculated for density and wrapped in PE bags. Before counting, the sample in air-tight plastic bag was kept for a period of 20 days to assure ^226^Ra in the samples to reach secular equilibrium with its daughter ^222^Rn [10,11,13]. The efficiency calibration of gamma-ray peaks was achieved by using S/N13000566 point source (the instrument equipped itself, Canberra Industries, Inc., Oak Ridge, TN, USA). The geometry of all counting samples was as the same as that of the standard samples. The counting time for all the samples was 24 h. The background counts (72 h) due to naturally occurring radionuclides in the environment around the detector were subtracted from that of each sample.

#### 2.2.2. ^90^Sr

^90^Sr (-^90^Y) was measured by a di- (2-ethylhexyl) phosphoric acid (HDEHP) extraction- β counting method according to the Technical Specification for Marine Radioactivity Monitoring [14]. The sample was counted using the gas-flow proportional alpha/beta counting system (LB4200, from Canberra Industries, Inc., Oak Ridge, TN, USA). The counting efficiency was measured by using the ^90^Sr-^90^Y standard reagent (National Institute of Metrology China, Beijing, China).

#### 2.2.3. Gross Beta

Gross beta determination is not an absolute but relative radioactivity-determination, which is also known as the beta index [15,16] because anthropogenic radionuclides are mostly beta-emitting. In the early warning and monitoring of nuclear weapon tests and nuclear accidents, it is used as one of the signal and degree index of radioactive pollution, and also the first screening item for the radiological hygiene inspection process. Gross beta was counted on a thin layer of fine ash on a planchette by using gas-flow proportional alpha/beta counting system (LB4200, from Canberra Industries, Inc., Oak Ridge, TN, USA). The counting efficiency was measured by using KCl (high purity reagent) (National Institute of Metrology China, Beijing, China) [12,17].

## 3. Results

Results are shown in Table 1, Figure 2, Figure 3 and Figure 4. The specific activities of ^238^U, ^228^Th, ^226^Ra, ^40^K and gross beta in the organisms were 1.01 ± 0.16 (<0.05–5.20), 2.46 ± 0.13 (0.30–14.50), 1.15 ± 0.06(0.11–3.58), 57.9 ± 2.53 (11.1–148.0) Bq/kg, respectively. Except twitch grass, ^228^Th and ^226^Ra were significantly below their limited concentrations in foods [18]. For anthropogenic radionuclides, the specific activities of ^137^Cs and ^90^Sr were (<0.003–0.088) and (0.021–0.316) mBq/kg, respectively. Other anthropogenic radionuclides of ^51^Cr, ^55^Fe, ^54^Mn, ^58^Co, ^60^Co, ^65^Zn and ^110m^Ag were below the minimum detectable activity, i.e., MDA. ^137^Cs and ^90^Sr were greatly lower than their limited concentrations in foods [18] and general action level [19] in the consumption food in China.

### 3.1. Terrestrial Organisms

#### 3.1.1. Unhulled Rice from Dongping and Dagou (Crops, *n* = 2)

In the unhulled rice from Dongping, ^238^U, ^226^Ra, ^228^Th, ^40^K and gross beta were higher than those from Dagou. Gross beta came mainly from the natural ^40^K contribution and was at the same order of magnitude as that in Daya bay in 1980s [20]. In the unhulled rice, both ^40^K and gross beta were higher than those in the rice (i.e., husked rice) in Daya bay (before and after Daya bay Nuclear power plants, i.e., GNPP) [21,22]. In the unhulled rice from Dagou, ^137^Cs and ^90^Sr were higher than those from Dongping. ^137^Cs was slightly higher than that in the rice (i.e., husked rice) investigated before the GNPP, but lower than that in the rice before the GNPP [21,22]. ^90^Sr was slightly higher than that in the rice (i.e., husked rice) in the past investigations [20,21,22].

#### 3.1.2. Sweet Potatoes from Dongping, Dagou and Huangjiangwei (Potatoes, *n* = 3)

The orders of the specific activities are as follows: ^238^U and ^226^Ra, Dongping > Dagou ≥ Huangjiangwei; ^228^Th, ^40^K and gross beta, Huangjiangwei > Dongping > Dagou; ^137^Cs, Dongping ≥ Dagou ≥ Huangjiangwei; ^90^Sr, Huangjiangwei > Dagou = Dongping. ^228^Th was the highest in Huangjiangwei because Huangjiangwei belongs to the HBRAYJB with a high-level thorium. Both ^40^K and gross beta were much similar to ^228^Th in three areas. ^40^K in Huangjiangwei was at the same level as that in Daya bay (1988–1990, Before the GNPP) [21], but was much lower than that in Dongping and Dagou. Both ^137^Cs and ^90^Sr were at the same order of magnitude as those in Day bay before the GNPP [21], but were lower than those in Sichuan province in 1985 [20].

#### 3.1.3. Turnip and Bok-choy and Sugarcane (Vegetables/Fruits)

*Turnip (n = 3)**from Dongping, Dagou and Huangjiangwei*—The orders of the specific activities are as follows: ^238^U, Huangjiangwei > Dagou and Dongping (below the MDA); ^226^Ra and ^228^Th, Huangjiangwei > Dongping > Dagou; ^40^K and gross beta, Dongping > Dagou > Huangjiangwei; ^137^Cs and ^90^Sr, Dagou > Dongping > Huangjiangwei. ^40^K was much higher than that in Daya bay (1988–1990, before GNPP) [21]. Gross beta was slightly higher than that in Daya bay before and after the GNPP [22]. ^137^Cs was at the same level as that before and after the GNPP [22,23]. ^90^Sr was slightly higher than that in Daya bay before the GNPP [22], but lower than those in Qinghai, Liaoning and Jiangsu provinces in China in 1985 [20].

*Bok-choy (n = 1) from Dongping*—Both ^40^K and gross beta were at the same level as those from Daya bay before and after the GNPP [21,22]. ^137^Cs was slightly higher than that in Daya bay before the GNPP, but at the same order of magnitude as those after the GNPP [22], and in Haiyang (2010–2012, before Haiyang NPP) (i.e., Haiyang NPP background (2011–2012) [24]. ^90^Sr was at the same level as that in Daya bay before GNPP [21,22], but was significantly lower than those in Ningxia, Shangdong and Sichan provinces in 1985 [20], in Daya bay after the GNPP [22] and in Haiyang (2010–2012, before Haiyang NPP) [24].

*Sugarcane (n = 1) from Dagou*—Gross beta was at the same order of magnitude as that in Daya bay in 1980s (7.4–89.1 Bq/kg, the mean value of 29.5 ± 0.6 Bq/kg) [20].

#### 3.1.4. Twitch Grass from Dagou, Dongping and Huangjiangwei (Fodder, *n* = 3)

The orders of the specific activities are as follows: ^238^U, ^40^K, gross beta and ^137^Cs, Dagou > Dongping ≥ Huangjiangwei; ^226^Ra, Dongping > Dagou > Huangjiangwei; ^228^Th, Dagou > Huangjiangwei > Dongping; ^90^Sr, Huangjiangwei ≥ Dongping > Dagou. Gross beta in this study was lower than that in Daya bay before GNPP (1985, 1987) [20]. ^137^Cs was at the same level as that in Daya bay after GNPP (1999–2003) [23], and lower than that in Haiyang (2011–2012, before NPP) [24]. ^90^Sr was significantly lower than that in Haiyang (2011–2012, before NPP) [24].

#### 3.1.5. Chicken and Freshwater Fish (Meat/Fish)

*Chicken (n = 1)**in**Dongping*—Gross beta was significantly higher than that in Daya bay before and after GNPP [22]. ^137^Cs in chicken from Dongping was higher than that from Daya bay before and after GNPP (1988 to 2003) [22,23]. ^90^Sr was at the same level as those in Daya bay before GNPP (1988–1990) [21] and after GNPP (1994–2001) [22]. However, ^90^Sr was significantly lower than that before GNPP [22], but higher than those (1990–2003) [23] in Daya bay.

*Freshwater fish (Tilapia) (n = 1)**in**Dongping*—Gross beta was at the same level as that in Daya bay before and after GNPP [22]. ^137^Cs in Dongping was lower than that in Daya bay after GNPP [22,23]. ^90^Sr was at the same level as that in Daya bay before and after GNPP (1990–2004) [21,22], but higher than the results (1990–2003) [23]. Meanwhile, ^90^Sr was significantly lower than that in bighead carp (0.095 Bq/kg) in Shangdong privence (1985) [25].

### 3.2. Marine Organisms

#### 3.2.1. Sea Fish (*n* = 4) from Dongping, Zhapo and Xiachuan

The orders of the specific activities as follows: ^238^U, Dongping > Zhapo >> Xiachuan (below the MDA); ^226^Ra and ^228^Th, ^40^K, gross beta and ^137^Cs, Dongping > Xiachuan > Zhapo. For two sorts of fishes (sea and freshwater), ^238^U, ^228^Th, ^226^Ra in sea fish were much lower than those in freshwater fish. ^4^^0^K and gross beta were slightly less than those in freshwater fish. ^228^Th and ^226^Ra in this study were significantly higher than those in west Daya bay (1993–2001) [26], and slightly lower than those in Daya bay (1997–1998) [27]. Furthermore, ^228^Th and ^226^Ra were at the same level as those in Bohai Sea and Yellow Sea (1970–1981) [28] and in Huludao, Lianyungang, Hangzhou bay [27]. ^238^U, ^40^K and gross beta in seafish were at the same level as those in the past investigations (1970–1981, 1993–2001 and 1997–1998) [26,27,28].

^137^Cs in sea fish was significantly higher than that freshwater fish (below the MDA). ^90^Sr was only analyzed in sea fish from Dongping in April 2012, 0.138 ± 0.008 Bq/kg. ^137^Cs was at the same level as that in Huludao, Lianyungang, Hangzhou bay and Daya bay (1997–1998) [27], but significantly lower than that in Bohai Sea and Yellow Sea (1970–1981) [28], in Daya bay before and after GNPP [23,26,29,30], and in Haiyang (2011–2012, before Haiyang NPP) [24]. In this study, ^90^Sr was higher than those in west Daya bay (1993–2001, after GNPP) [26] and in Haiyang (2011–2012, before Haiyang NPP) [24], but lower than those in Bohai Sea and Yellow Sea (1970–1981) [28] and in Daya bay before and after GNPP [29,30].

#### 3.2.2. Sea Shrimp from Dongping, Zhapo and Xiachuan (*n* = 5)

The orders of the specific activities are as follows: ^238^U, Dongping > Xiachuan > Zhapo; ^226^Ra and ^228^Th, Dongping >> Zhapo > Xiachuan; ^40^K and gross beta, Dongping ≥ Zhapo > Xiachuan. In this study, ^228^Th was significantly higher than those in Huludao, Lianyungang, Qingdao and Hangzhou bay (1997–1998) [27], in west Daya bay (1993–2001) [26], because ^228^Th in Dongping was an order of magnitude higher than others. Gross beta was significantly lower than those in Bohai Sea and Yellow Sea (1970–1981) [28], in west Daya bay (1993–2001) [26]. ^238^U, ^226^Ra and ^40^K were at the same level as those in the past investigations (1970–1981, 1993–2001 and 1997–1998) [26,27,28,31].

For sea shrimp samples, ^137^Cs in Zhapo was the highest, followed by in Xiachuan and in Dongping. ^90^Sr was only analyzed in sea fish from Dongping in April 2012 (0.243 ± 0.011 Bq/kg). ^137^Cs (*n* = 5) was at the same level as those in the past investigations [23,24,26,27,28,29,31]. ^90^Sr (*n* = 1) was significantly lower than those in Bohai Sea, Yellow Sea (1970–1981) and Fujian province (1982) [28,31], and at the same level as that in Haiyang (2011–2012, before Haiyang NPP) [24], while higher than that in west Daya bay (1993–2001) [26].

#### 3.2.3. Mollusks (e.g., Oysters, Mussels) from Dongping, Zhapo and Xiachuan (*n* = 5)

For the mollusics samples (*n* = 5, including oysters, mussels), the specific activities of ^238^U, ^228^Th and ^226^Ra, ^40^K and gross beta (*n* = 5) were 0.53 ± 0.06 (<0.12–0.93), 0.69 ± 0.03 (0.30–1.62), 0.66 ± 0.04 (0.11–2.37), 17.3 ± 0.7 (11.1–20.5) and 22.5 ± 2.0 (20.0–24.7) Bq/kg, respectively. For the oysters samples (*n* = 4), the specific activities of ^238^U, ^228^Th and ^226^Ra, ^40^K and gross beta were 0.40 ± 0.05 (<0.12–0.65), 0.69 ± 0.03 (0.30–0.64), 0.71 ± 0.04 (0.11–2.37), 18.0 ± 0.8 (13.7–20.5) and 22.4 ± 2.0 (20.0–24.7) Bq/kg, respectively. The orders of the specific activities are as follows: ^238^U, Xiachuan ≥ Zhapo > Dongping; ^226^Ra and ^228^Th, Dongping>> Xiachuan ≥ Zhapo; ^40^K and gross beta, Zhapo ≥ Dongping ≥ Xiachuan. In the study, ^238^U and ^228^Th in the mussels’ sample (*n* = 1) were higher than those in the oysters (*n* = 4), while ^40^K and gross beta were lower than those in the oysters. ^228^Th was significantly higher than that in Pearl hell in west Daya bay (1993–2001) [26], and significantly lower than those in shellfish in Huludao, Qingdao and Daya bay (1997–1998) [27]. ^238^U, ^226^Ra, ^40^K and gross beta were at the same level as those in the past investigations (1970–1981, 1993–2001 and 1997–1998) [26,27,28,31].

^137^Cs was only detected (0.020 ± 0.010 and 0.007 ± 0.002 Bq/kg) in the oysters from Dongping and Zhaopo. ^90^Sr was only analyzed in the oysters sample in Dongping (0.118 ± 0.005 Bq/kg). For the oysters samples (*n* = 4) and the mollusic samples (*n* = 5), ^137^Cs was slightly lower than those in the past investigations [24,27,28,29,31]; ^90^Sr (n = 1) was significantly higher than those in the past investigations [24,27,28,29,30,31].

## 4. Discussion

### 4.1. Differences and Comparisons of the Organismal Species

There were some differences between each radionuclide in various species organisms because of their habitats and environment. For natural U and Th series, ^228^Th belonging Th-series was generally higher than ^238^U and ^226^Ra in the same terrestrial specie organisms. The previous results suggested that there were two granite-hill areas with high background radiation (total area of ~540 km^2^), viz, Tongyou and Dong-anling (three villages of Yong’an, Huan’an and Liang’an). The high background radiation ought to originate from the weathered granite made up of fine monazite, which are washed continually and deposited into the surrounding basin region [1]. Radioactive thorium was constantly released from the fine granite rock under various geological process (tectonic process, water-rock processes and wind erosion), which will inevitably result in the kind of thorium-rich soil in Hongjiangwei. Many results also indicated that the HBRAYJ was mainly caused by the high concentration of ^228^Th in the hilly soils (the mean value of 206 ± 92 Bqkg^−1^) [2,32,33,34,35]. In soil in HBRAYJ, ^238^U, ^228^Th and ^226^Ra were three times as high as those in CA [36], while ^232^Th, ^228^Th and ^230^Th were generally 8.7, 7.7 and 9.2 times as high as those in CA, respectively [37,38,39]. In addition, ^222^Rn and ^220^Rn daughters were three times higher than those in CA and the mean concentration in China [40].

In Huangjiangwei, ^228^Th and ^90^Sr in sweet potato were significantly higher than those from Dagou and Dongping due to those sweet potato’s habits and environments. Sweet potato is a dicotyledonous plant, whose roots are vegetable. Sweet potato root can directly contact the soil and absorbs nutrients to accelerate the enrichment of thorium and calcium-like strontium. Furthermore, ^40^K in sweet potatoes also was higher than other terrestrial plants. The mean value of ^40^K spatial activities in sweet potato was 1.5 times as that in unhulled rice (*n* = 2), and about two times than those in turnip (*n* = 3), twitch grass (*n* = 3) and bok-choy (*n* = 1). Gross beta in the terrestrial organisms was greatly similar as ^40^K, because gross beta mainly contributes to natural ^40^K. Similarly, turnip is also root-vegetable so that ^228^Th was also the highest in turnip in Huangjiangwei.

^238^U, ^228^Th, ^226^Ra in Tilapia collected from Dagou were much higher than those in sea fish collected three areas (Dongping, Zhapo and Xiachuan). This may depend on Tilapia’s biological habits. Since Tilapia usually lives in freshwater as well as in salt water, i.e., river, lake, marsh and sea. Tilapia is a highly suitable fish for farming since it is fast growing and very tolerant in regards to water conditions (with less dissolved oxygen). Most Tilapia are omnivorous and feed on plants and debris in the water. Furthermore, ^40^K and gross beta were slightly less than those in freshwater fish. The mean value of ^90^Sr in seafish was 0.138 ± 0.008 Bq/kg, which was about one-second of that in freshwater fish.

^228^Th in the sea shrimp was greatly higher than that in sea fish and mollusic samples. Since sea shrimp belongs to bentonic organism in the sea bed, and can directly intake the plankton or benthic organisms in the seawater-sediment interface to accelerate the thorium enrichment. In sea shrimp in three areas (Dongping, Zhapo and Xiachuan), natural radionuclides-^238^U (0.63 ± 0.08 Bq/kg), ^228^Th (14.35 ± 0.36 Bq/kg), ^226^Ra (2.33 ± 0.08 Bq/kg) from Dongping were much higher than those from other two areas (Zhapo and Xiachuan). This may be closely related to river input into sea, including Moyang River and Shouchang river. Similarly, the aforementioned mechanism of HBRAYJ were also occurred. Those rich-thorium soil are washed by rain into rivers and eventually into the sea area near Doping. Thus, the uranium concentration (0.14–0.36μg/L, the mean value of 0.19 μg/L) in the water of Moyang River was much higher than that of other rivers in Guangdong Province [41]. That may be one reason that ^228^Th in the sea shrimp in our study was significantly higher than those in past investigations [26,27].

Natural radionuclides of ^238^U, ^228^Th, ^226^Ra and ^40^K in the organisms were originated from the surrounding soils. ^228^Th was significantly higher than ^238^U and ^226^Ra of natural U series in organisms due to the rich-Th soils in the HBRAYJ. For anthropogenic radionuclides, the YJNNP is not yet operational and could not have released anthropogenic pollutants. ^137^Cs and ^90^Sr should originated from global fallout. Even though our survey was conducted after in Marth 2011, we think that the effect from the FNA on the organisms in our study also very limited and negligible.

### 4.2. Radiological Risk Assessment

In the plants, the radionuclides are mainly derived from soil, water and some absorption through the leaves. Similarly, the radionuclides in the terrestrial and/or marine animals should originate from their ingested food/water and some absorption through the skin or respiratory systems [42,43]. The radioactive contamination in organisms depends on three main factors: (1) the duration of contact with the contaminated media, e.g., food and water, soil/bottom mud, atmosphere, (2) the organisms concentration ability and (3) the organisms’ maturity. We could not evaluate the influences of the first two factors because of the mobility and inter-individual differences of the organisms, and because of the lack of some data. Thus, it is difficult to determine the mechanisms underlying the contamination levels in various species. The internal doses from ingesting the organisms were easily assessed. Assuming that the entire consumption each year was solely of the species in this study, the committed effective internal doses from the ingestion is estimated by the following equation:(1)$$In-Dose=(\sum_{i}^{n}Dc_{i}A_{i})\cdot Ca\cdot Y$$
where the In-Dose is the internal doses from ingesting the organisms, Sievert (Sv); i is the radionuclide, e.g., ^238^U, ^226^Ra and so on; *Dc* is the radionuclide–specific committed effective dose coefficients for adult human ingestion, it converts the energy emitted from the ingested radioactivity into a radionuclide-specific, committed effective dose to adult humans, with units of Sv [44,45]; *Ca* is the annual per capita consumption rate, kg; *Y* is the percentage of the edible portion of the organism, %. As shown in Table 2, the assumed annual per capita consumption rates and the In-Dose for all types of species ranged from 1.81 to 136.49 μSv/a. For all organisms, the In-Dose from unhulled rice is the highest, yet the In-Dose from sugarcane is the lowest, less than one-fiftieth of that from unhulled rice. For the food chain between grass-cow-milk, we may assume that 250-kg cow can eat ~25 kg/day of twitch grass and then produce ~17.5 kg/day of milk in this study. If the conversion rate for each radionuclide is 100%, and the Ca for milk to humans is ~15 kg/a, the In-Dose is very low, only ~0.12 ± 0.01 nSv/a (0.15–0.22 nSv/a).

Figure 5 shows results of the assessed dose contributions of each radionuclide. The percentage of natural ^40^K-internal doses ranged from 16.8% to 67.5% (the mean value of 43.8%). In addition, ^226^Ra (from 16.9 to 59.1%, the mean values of 36.1%), ^228^Th (from 3.5% to 49.3%, the mean values of 15.5%) were another two main contributors. The internal doses from natural ^40^K were ~40 to ~1500 times as high as those from anthropogenic radionuclides (^137^Cs+^90^Sr).

The health risks resulting from a radiation dose are known for humans, but there are some large uncertainties to infer risks to health from such low dose at present [46]. Statistically significant elevations in cancer risk are observed at doses >100 mSv, and epidemiological studies cannot identify significant elevations in risk well below these levels [47]. For adult humans, the detriment-unadjusted nominal risk coefficient for fatal cancer is 4.1–4.8% per Sv of radiation dose [48]. Here, we consider additional cancer risk to humans due to the consumption. Even if humans consumed only the species in this study instead of other food for an entire year, the greatest increase in the probability of fatal cancer from anthropogenic radionuclides would be 0.000432% (0.000009–0.000653%). The effects of low levels of ionizing radiation on humans remain uncertainties, but our result shows clearly that these internal doses are very low. For the most exposed segments of human populations, this resulting cancer risks are far below levels that should cause concern.

## 5. Conclusions

Nuclear power is an effective way to achieve the goal of carbon dioxide emission reduction and realize the coordinated development of energy, economy and environment (3E system). At present, the YJNPP operation ensures energy needs and promotes sustainable economic development in Yangjiang and the adjacent areas. However, the YJNPP built in high-level areas is one of the few examples in the world. On the premise of ensuring the nuclear power and economic development, it is important to understand the radioactive level and the potential radiation risk systematically and scientifically. Our results suggested that the radioactive levels were too low to affect human health. There are differences in the concentrations of radionuclides in different species of organisms, even in the same species from various habitats. The distribution of natural radionuclides in the organisms were associated with the living environments. The anthropogenic radionuclides still originated from global fallout. Our results are useful for understanding the effects of low dose radiation on human in the HBRAYJ at present. It is useful to provide some basic data and assess the radiological risk from the HBRAYJ and YJNNP in future. In particular, it is also essential for understanding the effects of low dose radiation on non-human species in future.

## Figures and Tables

**Figure 1 ijerph-18-08767-f001:**
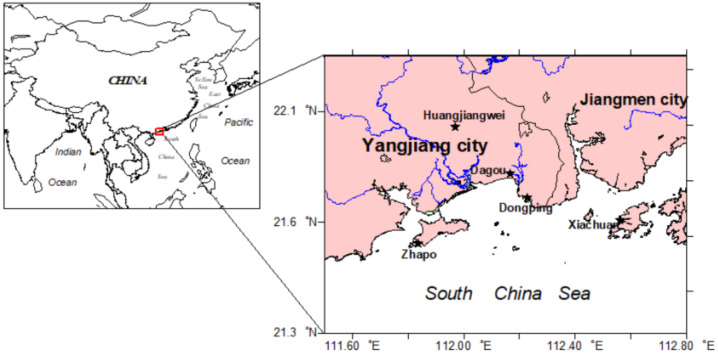
Sampling stations in Yangjiang and the adjacent areas.

**Figure 2 ijerph-18-08767-f002:**
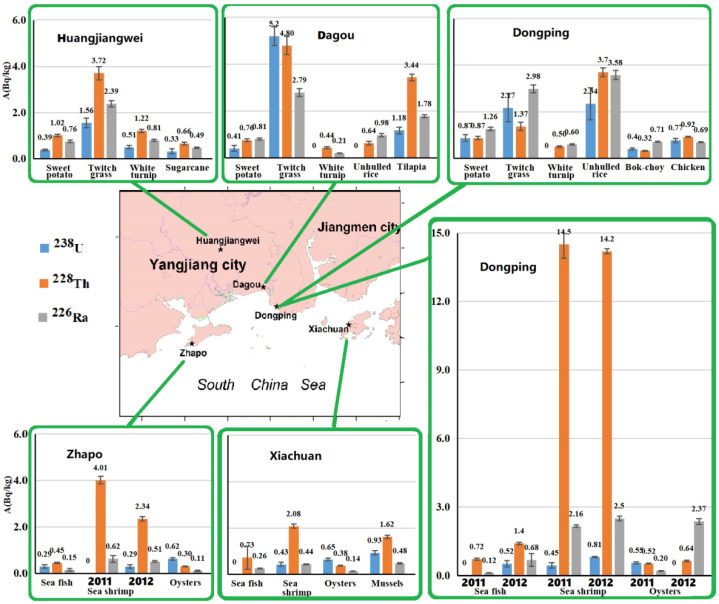
^238^U, ^228^Th and ^226^Ra in the organisms collected from Yangjiang and the adjacent areas.

**Figure 3 ijerph-18-08767-f003:**
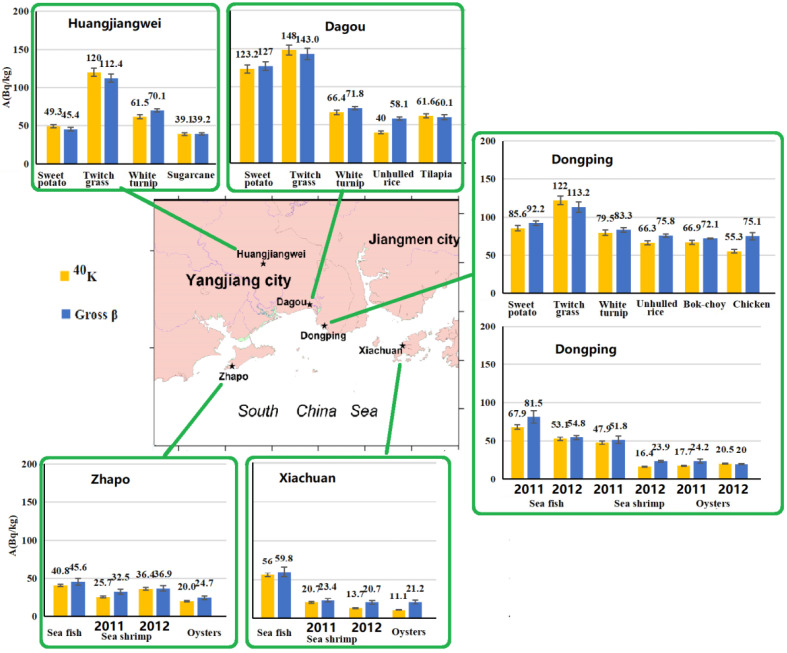
^40^K and gross beta in the organisms collected from Yangjiang and the adjacent areas.

**Figure 4 ijerph-18-08767-f004:**
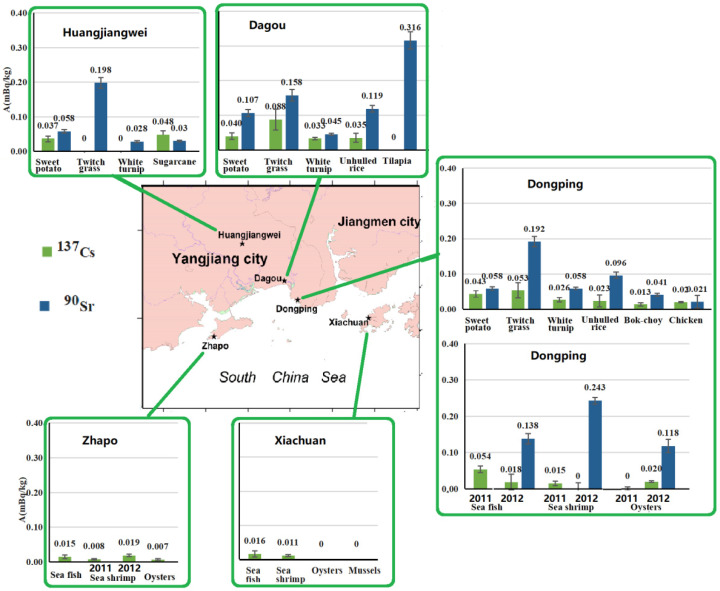
^137^Cs and ^90^Sr in the organisms collected from Yangjiang and the adjacent areas.

**Figure 5 ijerph-18-08767-f005:**
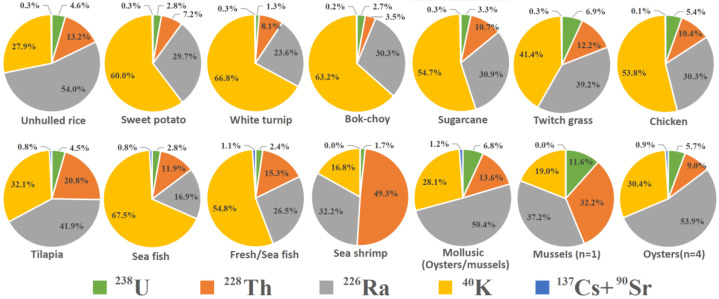
Dose contributions of each radionuclide.

**Table 1 ijerph-18-08767-t001:** ^238^U, ^228^Th, ^226^Ra, ^40^K, gross beta, ^137^Cs, ^90^Sr in the organisms from Yangjiang and the adjacent areas.

Species	Sorts	Date	Natural Radionuclides				Gross Beta	Anthropogenic Radionuclides	
		M/D/Y	^238^U	^228^Th	^226^Ra	^40^K		^137^Cs	^90^Sr
Terrestrial organisms from Dongping, Yangjiang city									
Unhulled rice	Crops	12/3/2012	2.34 ± 0.68	3.70 ± 0.32	3.58 ± 0.19	66.3 ± 2.8	75.8 ± 2.3	0.023 ± 0.017	0.096 ± 0.009
Sweet potato	Potatoes	12/3/2012	0.87 ± 0.14	0.87 ± 0.06	1.26 ± 0.07	85.6 ± 3.8	92.2 ± 3.1	0.043 ± 0.009	0.058 ± 0.005
Bok-choy	Vegetables/fruits	12/3/2012	0.40 ± 0.05	0.32 ± 0.02	0.71 ± 0.02	66.9 ± 2.8	72.1 ± 0.6	0.013 ± 0.005	0.041 ± 0.004
Turnip		12/3/2012	<0.09	0.50 ± 0.03	0.60 ± 0.03	79.5 ± 3.5	83.3 ± 3.2	0.026 ± 0.006	0.058 ± 0.004
Twitch grass	Fodder	12/3/2012	2.17 ± 0.60	1.37 ± 0.17	2.98 ± 0.17	122 ± 5.6	113.2 ± 6.7	0.053 ± 0.022	0.192 ± 0.014
Chicken	Meat/fish/shrimp	12/3/2012	0.77 ± 0.09	0.92 ± 0.04	0.69 ± 0.02	55.3 ± 2.3	75.1 ± 4.7	0.020 ± 0.002	0.021 ± 0.018
Marine organisms from Dongping, Yangjiang city									
Sea fish	Meat/fish /shrimp	8/5/2011	<0.38	0.72 ± 0.04	0.12 ± 0.01	67.9 ± 2.8	81.5 ± 8.2	0.054 ± 0.008	-
		4/23/2012	0.52 ± 0.13	1.40 ± 0.05	0.68 ± 0.28	53.1 ± 2.2	54.8 ± 2.7	0.018 ± 0.003	0.138 ± 0.008
Sea shrimp		8/5/2011	0.45 ± 0.12	14.5 ± 0.6	2.16 ± 0.05	47.9 ± 2.0	51.8 ± 5.2	0.015 ± 0.005	-
		8/23/2012	0.81 ± 0.03	14.2 ± 0.11	2.50 ± 0.10	16.4 ± 0.7	23.9 ± 1.2	<0.005	0.243 ± 0.011
Oysters		8/5/2011	0.55 ± 0.05	0.52 ± 0.02	0.20 ± 0.01	17.7 ± 0.7	24.2 ± 2.5	<0.004	-
		4/23/2012	<0.12	0.64 ± 0.04	2.37 ± 0.13	20.5 ± 0.9	20.0 ± 1.0	0.020 ± 0.010	0.118 ± 0.005
Terrestrial organisms sampled from Dagou, Yangjiang city									
Unhulled rice	Crops	12/3/2012	<0.20	0.64 ± 0.08	0.98 ± 0.07	40.0 ± 1.8	58.1 ± 1.9	0.035 ± 0.013	0.119 ± 0.009
Sweet potato	Potatoes	12/3/2012	0.41 ± 0.12	0.76 ± 0.06	0.81 ± 0.05	123.2 ± 5.5	127 ± 5.4	0.040 ± 0.009	0.107 ± 0.010
Turnip	Vegetables/fruits	12/3/2012	<0.05	0.44 ± 0.04	0.21 ± 0.01	66.4 ± 3.0	71.8 ± 2.1	0.033 ± 0.003	0.045 ± 0.003
Twitch grass	Fodder	12/3/2012	5.20 ± 0.40	4.80 ± 0.39	2.79 ± 0.17	148 ± 6.7	143.0 ± 7.2	0.088 ± 0.030	0.158 ± 0.016
*Tilapia*	Meat/fish/shrimp	12/3/2012	1.18 ± 0.14	3.44 ± 0.14	1.78 ± 0.06	61.6 ± 2.6	60.1 ± 3.1	<0.033	0.316 ± 0.025
Organisms sampled from Huangjiangwei, Yangjiang city									
Sweet potato	Potatoes	12/3/2012	0.39 ± 0.04	1.02 ± 0.05	0.76 ± 0.05	49.3 ± 2.2	45.4 ± 2.3	0.037 ± 0.007	0.058 ± 0.005
Turnip	Vegetables/fruits	12/3/2012	0.51 ± 0.07	1.22 ± 0.07	0.81 ± 0.04	61.5 ± 2.7	70.1 ± 2.0	<0.003	0.028 ± 0.003
Sugarcane		12/3/2012	0.33 ± 0.11	0.66 ± 0.06	0.49 ± 0.03	39.1 ± 1.8	39.2 ± 1.5	0.048 ± 0.011	0.030 ± 0.002
Twitch grass	Fodder	12/3/2012	1.56 ± 0.21	3.72 ± 0.29	2.39 ± 0.14	120 ± 5.4	112.4 ± 5.6	<0.013	0.198 ± 0.15
Marine organisms sampled from Zhapo, Yangjiang city									
Sea fish	Meat/fish/shrimp	8/5/2011	0.29 ± 0.07	0.45 ± 0.03	0.15 ± 0.01	40.8 ± 1.7	45.6 ± 4.6	0.015 ± 0.005	-
Sea shrimp		8/5/2011	<0.20	4.01 ± 0.16	0.62 ± 0.02	25.7 ± 1.1	32.5 ± 3.3	0.008 ± 0.002	-
		8/24/2012	0.29 ± 0.07	2.34 ± 0.10	0.51 ± 0.02	36.4 ± 1.5	36.9 ± 3.7	0.019 ± 0.003	-
Oysters		8/5/2011	0.62 ± 0.05	0.30 ± 0.02	0.11 ± 0.01	20.0 ± 0.8	24.7 ± 2.5	0.007 ± 0.002	-
Marine organisms sampled from Xiachuan, Jiangmen city									
Sea fish	Meat/fish/shrimp	8/5/2011	<0.0.35	0.73 ± 0.5	0.26 ± 0.02	56.0 ± 2.4	59.8 ± 6.0	0.016 ± 0.009	-
Sea shrimp		8/5/2011	0.43 ± 0.09	2.08 ± 0.09	0.44 ± 0.02	20.7 ± 0.9	23.4 ± 2.4	0.011 ± 0.003	-
Oysters		8/5/2011	0.65 ± 0.05	0.38 ± 0.02	0.14 ± 0.01	13.7 ± 0.6	20.7 ± 2.0	<0.007	-
Mussels		4/24/2012	0.93 ± 0.09	1.62 ± 0.07	0.48 ± 0.02	11.1 ± 0.5	21.2 ± 2.1	<0.014	-

Note: “-” represents the sample was not analyzed. ^238^U, ^228^Th, ^226^Ra, ^40^K, ^137^Cs, ^90^Sr and gross β activity are in the unit of Bq/kg.

**Table 2 ijerph-18-08767-t002:** Committed effective doses from ingesting food to humans.

Food Sorts	^238^U (Bq/kg)	^228^Th (Bq/kg)	^226^Ra (Bq/kg)	^40^K (Bq/kg)	^137^Cs (Bq/kg)	^90^Sr (Bq/kg)	Ca ^b^ (kg/a)	Y (%)	In-Dose ^d^ (μSv/a)
D_C_ (nSv/Bq) ^a^	45.0	72.0	280	6.2	13.0	28.0			
Unhulled rice (*n* = 3)	<9.00–105.30 53.78 ± 30.6	46.08–266.40 156.24 ± 14.40	274.40–1002.40 638.4 ± 36.4	248.0–411.1 329.5 ± 14.3	0.299–0.455 00.377 ± 0.195	2.688–3.332 3.010 ± 0.252	112.2	68	43.77–136.49 90.13 ± 7.33
Sweet Potatoes (*n* = 3)	17.55–39.15 25.05 ± 4.50	54.72–73.44 63.60 ± 4.08	212.80–352.80 264.13 ± 15.87	305.7–763.8 533.4 ± 23.8	0.481–0.559 0.520 ± 0.108	1.624–2.996 2.081 ± 0.187	47.6	100	28.22–58.68 42.31 ± 2.31
Turnip (*n* = 3)	<2.25–22.95 8.18 ± 3.15	31.68–87.84 51.84 ± 3.36	58.80–226.80 151.20 ± 7.47	381.3–492.9 428.6 ± 19.0	<0.039–0.429 0.260 ± 0.059	0.784–1.624 1.223 ± 0.093	38.7	100	18.32–32.22 24.82 ± 1.28
Bok-choy (*n* = 1)	18.00 ± 2.25	23.04 ± 1.44	198.80 ± 5.60	414.78 ± 17.36	0.17 ± 0.07	1.15 ± 0.11	38.7	100	25.38 ± 1.04
Chicken (*n* = 1)	34.65 ± 4.05	66.24 ± 2.88	193.20 ± 5.60	342.86 ± 14.26	0.26 ± 0.03	0.59 ± 0.50	10	60	3.83 ± 0.16
Freshwater Fish (*n* = 1) Tilapia	53.10 ± 6.30	247.68 ± 10.08	498.40 ± 16.80	381.92 ± 16.12	<0.429	8.85 ± 0.70	36	60	25.70 ± 1.08
Sea Fish (*n* = 4)	13.05–23.40 14.12 ± 4.50	32.40–100.80 59.40 ± 11.16	33.60–190.40 84.70 ± 22.40	253.0–421.0 337.6 ± 14.1	0.195–0.702 0.335 ± 0.081	3.864 ± 0.224	36	60	7.26–15.99 10.80 ± 1.13
Sea/freshwater Fish (*n* = 5)	13.05–53.10 14.87 ± 5.10	32.40–247.68 97.06 ± 10.94	33.60–498.40 167.44 ± 21.28	253.0–421.0 346.5 ± 14.5	<0.429–0.702 0.289 ± 0.081	3.864–8.848 6.356 ± 0.462	36	60	7.26–26.56 13.66 ± 1.13
Seashrimp (*n* = 5)	9.00–36.45 18.27 ± 3.49	149.76–1044.0 534.67 ± 15.26	123.20–700.00 348.88 ± 11.76	101.7–297.0 182.4 ± 7.7	0.104–0.247 0.141 ± 0.042	6.804 ± 0.308	18	60	4.22–22.44 11.71 ± 0.41
Oysters (*n* = 4)	24.75–29.25 20.81 ± 2.25	21.60–46.08 33.12 ± 1.80	30.80–663.60 197.40 ± 11.20	109.7–127.1 111.4 ± 4.7	0.091–0.260 0.106 ± 0.078	3.304 ± 0.140	18	100	1.79–9.36 3.95 ± 0.22
Mussels (*n* = 1)	41.85 ± 4.05	116.64 ± 5.04	134.40 ± 5.60	68.82 ± 3.10	<0.05	/ ^c^	18	100	3.91 ± 0.19
Mollusic(Oysters/mussels) (*n* = 5)	24.75–41.85 25.02 ± 2.70	21.60–116.64 49.82 ± 2.45	30.80–663.60 184.80 ± 10.08	68.8–127.1 102.9 ± 4.3	0.091–0.260 0.945 ± 0.07	3.304 ± 0.140	18	100	1.61–10.29 3.96 ± 0.21
Sugarcane(*n* = 1)	14.85 ± 4.95	47.52 ± 4.32	137.20 ± 8.40	242.42 ± 11.16	0.62 ± 0.14	0.84 ± 0.06	5.1	80	1.81 ± 0.12
Twitchgrass (*n* = 3)	70.20–234.00 133.95 ± 18.15	98.64–345.60 237.36 ± 20.40	669.20–834.40 761.60 ± 44.80	744.0–917.6 806.0 ± 36.6	0.689–1.144 0.917 ± 0.338	4.424–5.544 5.115 ± 0.420	- ^e^	- ^e^	0.00015–0.00022 0.00018 ± 0.00001

Note: ^a^ Dcradionuclide–specific committed effective dose coefficients for adult human ingestion [45]. ^b^ the annual per capita consumption rate (i.e., Ca) are for all types of species combined, whereas the dose calculations conservatively assumed the entire consumption consisted solely of each species. Assumed exposure time is 50a for adults. ^c^ Lack of data. ^d^ In-Dose, the internal doses from ingesting the food to humans from the personal annual consumption assessed. ^e^ the method estimated see Section 4.2.

## Data Availability

The data presented in this study are available in insert article here.

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
