# Peer review of "Assessment of Radioactivity Level in the Terrestrial and Marine Organisms in Yangjiang and Its Adjacent Areas (China)"

_ijerph, 2021, doi:10.3390/ijerph18168767_

Round 1

Reviewer 1 Report

Dose contributions of each radionuclide chart should be reviewed because the quality of representation is not very good.

Author Response

R:OK.

It is known that there are many methods for dose calculation. Here, we only choose the method recommended by ICRP(1996, 2012) . The histogram ( or Bar chart ) of dose estimation results of each radionuclide in this paper is not obvious enough. In order to represent the contribution of radionuclides more clearly, Bar chart of Figure 5 is changged in Pie chart about dose contributions of each radionuclide.

Reviewer 2 Report

  The abstract does not present results, it must be substantially improved. The publication has a great weakness which is the introduction. This does not contextualize the problem, does not provide complete previous information, both about the locations where the evaluation is carried out and especially about the scientific background involved in this work, which is very important given the complexity of the subject. Therefore, the reading of the later chapters is very difficult. The presentation of the graphic information is not clear and makes it difficult to read and understand, I would recommend taking the graphics from the maps and linking them in a more pedagogical way. A large amount of data is found, but it must be presented in a more orderly and friendly way. The conclusions must be pointed out and distinguish one from the other. Finally, it is rescued that it is a work with a lot of information, on a subject with a huge environmental and public health impact that it is necessary to expose.

Author Response

Reviewer #2:

Comments and Suggestions for Authors

  • The abstract does not present results, it must be substantially improved.

R:OK.

The original abstract In order to assess the radioactive level in the terrestrial and marine organisms in Yangjiang and the adjacent areas, 90Sr, gross beta, and gamma-emitting radionuclides (238U, 226Ra, 228Th, 226Ra, 40K, 137Cs, 51Cr, 55Fe, 54Mn, 58Co, 60Co and 65Zn ) were analyzed from 2011 to 2012. The annual effective doses was estimated in the high natural radioactive background areas in Yangjiang (HBRAYJ). Results show that 228Th was significantly higher than 238U and 226Ra in organisms; 51Cr, 55Fe, 54Mn, 58Co, 60Co, 65Zn and 110mAg were below the minimum detectable activity; 228Th, 226Ra, 40K, 137Cs and 90Sr were greatly lower than the limited concentrations in Chinese foods. The internal dose mainly contributes to natural 40K, 226Ra and 228Th. It is useful to provide some basic data and assess the radiological risk from the HBRAYJ and Yangjiang nuclear power plants in future.”is revised in order to present results as follows:

“In order to assess the radioactive level in the terrestrial and marine organisms in Yangjiang and the adjacent areas, 90Sr, gross beta, and gamma-emitting radionuclides (238U, 226Ra, 228Th, 226Ra, 40K, 137Cs, 51Cr, 55Fe, 54Mn, 58Co, 60Co and 65Zn ) were analyzed from 2011 to 2012. The annual effective doses was estimated in the high natural radioactive background areas in Yangjiang (HBRAYJ). The specfic activities of 238U, 228Th, 226Ra, 40K and 137Cs in all organisms were <0.05-5.20, 0.30-14.50, 0.11-3.58, 11.1-148.0, and <0.003-0.088 Bq/kg, whilst 51Cr, 55Fe, 54Mn, 58Co, 60Co, 65Zn and 110mAg were below the minimum detectable activity. 90Sr and gross beta special activities were 20.0-143.0 and 0.021-0.316 Bq/kg. Results show that 228Th was significantly higher than 238U and 226Ra of natural U series in organisms due to the rich-Th soils in the HBRAYJ; 228Th, 226Ra, 40K, 137Cs and 90Sr were greatly lower than the limited concentrations in Chinese foods. The internal dose mainly contributes to natural 40K, 226Ra and 228Th. It is useful to provide some basic data and assess the radiological risk from the HBRAYJ and Yangjiang nuclear power plants in future.”

  • The publication has a great weakness which is the introduction. This does not contextualize the problem, does not provide complete previous information, both about the locations where the evaluation is carried out and especially about the scientific background involved in this work, which is very important given the complexity of the subject.Therefore, the reading of the later chapters is very difficult.

R:OK.

The introduction has been modified.

  • In order to contextualize the problem and provide complete previous information, Paragraph2 is added at the end of Paragraph 1 as follows:It is known that nuclear power is a clean and high-quality energy that can will improve China's energy supply structure, ensure energy and economic security and protect the environment.The development of Yangjiang city was always restricted by power and infrastructure at past.The YJNPP is a nuclear power plant built within/near the HBRAYJ. Such examples of building nuclear power plants in high-level areas are rare in the world. It is a necessary task to investigate systematically the current situation of radioactivity and assess scientifically the potential risk near the YJNPP near the HBRAYJ.
  • In order to contextualize the problem further,at the end of Paragraph 3 as follows: Our dose calculations should help the public quantitatively assessing the risk associated with consumption of these popular organisms. Just our survey was conducted before and after the YJNNP operation, so it is expected that the results is useful to the routine and emergency radioactive monitoring around the YJNNP.

After the modification, the information about both the locations and the scientific background should be clear in this paper.

  • The presentation of the graphic information is not clear and makes it difficult to read and understand, I would recommend taking the graphics from the maps and linking them in a more pedagogical way. A large amount of data is found, but it must be presented in a more orderly and friendly way.

R:OK.

  • Fig 2,3 and Fig 4 have been modifiedto make the graphic information  In the process of revising the Fig. 2, 3 and 4, we tried to taking the graphics from the maps, but the effect shown in the figure is not very good. So we redrawn Fig. 2, 3 and 4 to make presentation of the graphic information clear. Morever, the introduction of maps is helpful to reveal the regional distribution characteristics of radionuclides, and to understand the source of natural radioactivity. 
  • In order to represent the quality well, Bar chart of Figure 5 is changged in Pie chart about dose contributions of each radionuclide.

  • The conclusions must be pointed out and distinguish one from the other. Finally, it is rescued that it is a work with a lot of information, on a subject with a huge environmental and public health impact that it is necessary to expose.

R: OK.

The “Conclusions” has been modified as follows:

Nuclear power is an effective way to achieve the goal of carbon dioxide emission reduction and realize the coordinated development of energy, economy and environment (3E system). At present, the YJNPP operation ensures energy needs and promotes sustainable economic development in Yangjiang and the adjacent areas. Hoever, the YJNPP built in high-level areas is one of the few examples in the world. On the premise of ensuring the development of nuclear power and economic development , it is impotant to understand the radioactive level and the potential radiation risk systematically and scientifically. Because the effects of low dose radiation on human in the HBRAYJ has been still an unsolved but very important issue. Also, the effects of low dose radiation on non-human species has become a concern since the beginning of this century. Here, some natural and anthropogenic radionuclides and thier radiation dose were analyzed in the terrestrial and marine organisms in Yangjiang and the adjacent areas (including the HBRAYJ). Our results are useful for understanding the effects of low dose radiation on human in the HBRAYJ at present. It is useful to provide some basic data and assess the radiological risk from the HBRAYJ and Yangjiang nuclear power plants in future. In particular, it is also essential for understanding the effects of low dose radiation on non-human species in future.

Reviewer 3 Report

The authors report both natural and anthropogenic radioactivity  in terrestrial plants and animals and in aquatic and marine animals from Southeastern coastal mainland China.  The period of collection was over two years from 2011 and 2012 soon after the Fukishima disaster.

Beyond the need for an English based copy edit as there are many flaws in usage, tense and grammar as well as sentence construction that must be addressed, the following is suggested: 

The authors should emphasize fission products such as cesium 137 and strontium 90 in their discussion and conclusions. This reviewer appreciates all the work done on naturally occurring radionuclides however in the biological context the radioactivity from these sources have been around for at least a billion years.

Although they provide a baseline of radioactivity for the region, the primary contribution of this manuscript is the source and extent of radioactivity contributed from nuclear power plants. This was only mentioned briefly in the abstract and only as a rationale for the study.

The authors should revise their results, figures and tables to emphasize the impact from man-made radioactivity. 

Beta counts were also conducted and reported. Please provide an interpretation for the results.

The work is significant , given the level of development of nuclear power reactors in this region of China. Moreover it does provide baseline data that may help understand ongoing and future nuclear incidents .

Author Response

Reviewer #3:

Comments and Suggestions for Authors

1.The authors report both natural and anthropogenic radioactivity  in terrestrial plants and animals and in aquatic and marine animals from Southeastern coastal mainland China. The period of collection was over two years from 2011 and 2012 soon after the Fukishima disaster. Beyond the need for an English based copy edit as there are many flaws in usage, tense and grammar as well as sentence construction that must be addressed, the following is suggested: 

R: OK.

An English based copy edit as there are many flaws in usage, tense and grammar as well as sentence construction that must be addressed,

  • In Page 1/14, line 43-44, “,” has been changed into “e.g.,”. Similarly, all “eg.” have been changed in other place in this paper.
  • Paragraph 1: “Yangjiang nuclear power plants(i.e., YJNPP) base” has been changed into “Yangjiang nuclear power plants base (i.e., YJNPP)”;
  • Paragraph 1: the last sentence“Some radionuclides (eg., 3H in water and gamma-emittingin marine sediments) and radiation effect had been studied near YJNPP (Zhang et al., 2009; Wu et al., 2018; Deng et al., 2015). ” has been changed into “Some radionuclides (eg., 3H in water and gamma-emitting radionuclides in marine sediments) and radiation effect have been studied near YJNPP (Zhang et al., 2009; Wu et al., 2018; Deng et al., 2015). ”;
  • In Table 1. and Table 2., the Font size of words is reduced.
  • In the Table 2., note is transfer to below the table.
  • In Page 4/14, line 120, “the special activity” and “the special activities” have been changed to “the specific activity” and “the specific activities” . Similarly, all “the special activity” have been changed in other place in this paper.
  • In Page 9/14, line 316”natural radionuclides-238U(0.63±0.08), 228Th (14.35±0.36), 226Ra (33±0.08 ) from ”  has been changed into “natural radionuclides-238U (0.63±0.08 Bq/kg), 228Th (14.35±0.36 Bq/kg), 226Ra (2.33±0.08 Bq/kg) from” .
  1. The authors should emphasize fission products such as cesium 137 and strontium 90 in their discussion and conclusions. This reviewer appreciates all the work done on naturally occurring radionuclides however in the biological context the radioactivity from these sources have been around for at least a billion years.Although they provide a baseline of radioactivity for the region, the primary contribution of this manuscript is the source and extent of radioactivity contributed from nuclear power plants. This was only mentioned briefly in the abstract and only as a rationale for the study.

R: OK.

Such examples, including the YJNPP, of building nuclear power plants in high-level areas are rare in the world. I think that it is a necessary task to investigate systematically the current situation of radioactivity including natrual and anthropogenic radionuclides. Morever, in order to assess scientifically the potential risk near the YJNPP near the HBRAYJ, our dose calculations include natrual and anthropogenic radionuclides. It should help the public quantitatively assessing the risk associated with consumption of these popular organisms. Just our survey was conducted before and after the YJNNP operation, so it is expected that the results is useful to the routine and emergency radioactive monitoring around the YJNNP.

In this paper, natrual and anthropogenic radionuclides is re-emphasized here in Introduction, Resluts, and Conclusions, tables.

  1. The authors should revise their results, figures and tables to emphasize the impact from man-made radioactivity. 
    R:OK.

 1) In Page 4/14, line 124-128, in order to emphasize man-made radioactivity, their results were redescribed as follows: Results are shown in Table 1, Figure 2, Figure 3 and Figure 4. The specific activities of 238U, 228Th, 226Ra, 40K, and gross beta in the organisms were 1.01±0.16 (<0.05-5.20), 2.46±0.13(0.30-14.50), 1.15±0.06(0.11-3.58), 57.9±2.53 (11.1-148.0) Bq/kg, respectively. Except twitch grass, 228Th and 226Ra were significantly below their limited concentrations in foods (GB 14882-1994). For anthropogenic radionuclides, the specific activities of 137Cs and 90Sr were (<0.003-0.088) and (0.021-0.316) mBq/kg, respectively. Other anthropogenic radionuclides of 51Cr, 55Fe, 54Mn, 58Co, 60Co, 65Zn and 110mAg were below the minimum detectable activity, i.e., MDA. 137Cs and 90Sr were greatly lower than their limited concentrations in foods (GB 14882-1994) and general action level (GB 18871-2002) in the consumption food in China.

  2)  In Page 3/14 line 119- Page 3/14 line 120, in Table 1., the header content is modified to indicate that 238U, 228Th, 226Ra and 40K are natural radionuclides, and 137Cs and 90Sr are anthropogenic radionuclides.

3) Fig.2,3 and 4 are modified clearly.

  1. Beta counts were also conducted and reported. Please provide an interpretation for the results.

R: OK.

Gross beta determination is not an absolute but relative radioactivity-determination, which is also known as the beta index because anthropogenic radionuclides are mostly beta-emitting. The detection of gross beta is simple and fast, and is always one of the preferred methods for radioactive monitoring. In the early warning and monitoring of nuclear weapon tests and nuclear accidents, it is used as the signal and degree index of radioactive pollution, and it is also the first screening item for the radiological hygiene inspection process. Gross beta is expected to provide some basic data and help to assess the radiological risk.

In Page 3/14, line 110-116, the sentences has been changed into “Gross beta determination is not an absolute but relative radioactivity-determination, which is also known as the beta index (Rice et al., 2012; Zhao et al., 2013) because anthropogenic radionuclides are mostly beta-emitting. In the early warning and monitoring of nuclear weapon tests and nuclear accidents, it is not only one of the signal and degree indicators of radioactive pollution, but also the first screening itemfor the radiological hygiene inspection process. Gross beta was counted on a thin layer of fine ash on a planchette by using gas-flow proportional alpha/beta counting system (LB4200, from Canberra U.S.). The counting efficiency was measured by using KCl (high purity reagent) (Nationalof Metrology P.R.China) (Zhou et al., 2015, 2018).

6.The work is significant , given the level of development of nuclear power reactors in this region of China. Moreover it does provide baseline data that may help understand ongoing and future nuclear incidents .

R:OK.

We have emphasized the significance of the research many times in this paper, as mentioned above.

Round 2

Reviewer 2 Report

Review Report 2

The abstract was improved.

It was improved the introduction, but does not provide information, about the scientific background involved in this work, which is very important given the complexity of the subject, does not explain the relationship between nuclear plant and the contamination of radioactive elements, does not explain the mechanisms of transfer of radioactive pollutants to living organisms.

  I reiterate, the presentation of the graphic information is not clear and makes it difficult to read and understand, I would recommend keep the graphics from the maps and replicate them in the text in a more readable size, zoom type, and reference them to the respective map

The conclusions have paragraphs that are typical of a discussion, the conclusions must be refined and clarified, ultimately making them relevant.

Author Response

Review Report 2

  1. The abstract was improved.It was improved the introduction, but does not provide information, about the scientific background involved in this work, which is very important given the complexity of the subject, does not explain the relationship between nuclear plant and the contamination of radioactive elements, does not explain the mechanisms of transfer of radioactive pollutants to living organisms.

R: OK.

In order to explain the relationship between nuclear plant and the contamination of radioactive elements, the introduction has been changed as follows:

  • Page 1/14,Line 37-38, “where six nuclear power units (6×1080 MW) will been built in future”has been changed into “where six nuclear power units (6×1080 MW) havebeen built at present.”
  • Page 1/14,Line 41-42,“Five nuclear power units hadbeen put into commercial operation until May 25, ”has been changed into “At present, sixnuclear power units had been put into commercial operation until July 24, 2019.”
  • Page 1/14,Line53-55,”...rare in the world. It is a necessary task to investigate systematically the current situation of radioactivity and assess scientifically the potentialrisk near the YJNPP near the HBRAY.”has been changed into

 “...are in the world. The effects of low dose radiation on human in the HBRAYJ has been still an unsolved but very important issue. Also, the effects of low dose radiation on non-human species has become a concern since the beginning of this century. So it is a necessary task to investigate systematically the current radioactive levels and assess scientifically the potential risk.”

  • Page 2/14,Line 59-70, ”Here,some organisms were collected according to the local production and diet habits in Yangjiang and the adjacent areas. This radioactive investigation in various species organisms is firstly and systematically conducted before and after the YJNPP commercial operation. Some items of 238U, 228Th, 226Ra, 40K, 90Sr, 137Cs, 134Cs, 51Cr, 55Fe, 54Mn, 58Co, 60Co, 65Zn, 110mAg and gross beta were analyzed to estimate annual effective dose near the HBRAYJ and YJNPP. Our dose calculations should help the public quantitatively assessing the risk associated with consumption of these popular organisms. Just our survey was conducted before and after the YJNNP operation, so it is expected that the results is useful to the routine and emergency radioactive monitoring around the YJNNP. This study is also expected to provide some basic data and help to assess the radiological risk from the HBRAYJ and Yangjiang nuclear power plants in ” has been changed in to

 “Here, some organisms were collected in Yangjiang and the adjacent areas during August 2011- April 2012, according to the local production and diet habits. This radioactive investigation in various species organisms is conducted firstly and systematically. Some items of 238U, 228Th, 226Ra, 40K, 90Sr, 137Cs, 134Cs, 51Cr, 55Fe, 54Mn, 58Co, 60Co, 65Zn, 110mAg and gross beta were analyzed to estimate annual effective dose near the HBRAYJ and YJNPP. Just sampling time is after the Fukushima Nuclear Power Plants Accident (FNA) in Marth 2011, and before the YJNPP commercial operation. So our results are expected to provide some basic background data. The dose calculations should help the public quantitatively assessing the risk associated with consumption of these popular organisms. It is useful to the routine and emergency radioactive monitoring around the YJNNP. This study is also expected and help to assess the radiological risk from the HBRAYJ and Yangjiang nuclear power plants in future.”    

  • Morever, in order to echo the issues inthe introductionand explain furtherly the relationship between nuclear plant and the contamination of radioactive elements, in Page 10/14,Line 343-344,another paragraph was added as follows:

Natual radionuclides of 238U, 228Th, 226Ra, and 40K in the organisms were originated from the surrounding soils. 228Th was significantly higher than 238U and 226Ra of natural U series in organisms due to the rich-Th soils in the HBRAYJ. For anthropogenic radionuclides, the YJNNP is not yet operational and could not have released anthropogenic pollutants. 137Cs and 90Sr should originated from global fallout. Althrough our survey was conducted after in Marth 2011, we think that The effect from the Fukushiama nuclear accident (FNA) on the organisms in our study also very limited and negliglible. 

  1. I reiterate, the presentation of the graphic information is not clear and makes it difficult to read and understand, I would recommend keep the graphics from the maps and replicate them in the text in a more readable size, zoom type, and reference them to the respective map

R: OK.

Figure 2, 3, 4 had been modified, or re-conceived to represent the graphic theme and provide sufficient information to the readers. 

  1. The conclusions have paragraphs thatare typical of a discussion, the conclusions must be refined and clarified, ultimately making them relevant.

R: OK.

The conclusion has been modified without some discussions as follows:

Nuclear power is an effective way to achieve the goal of carbon dioxide emission reduction and realize the coordinated development of energy, economy and environment (3E system). At present, the YJNPP operation ensures energy needs and promotes sustainable economic development in Yangjiang and the adjacent areas. Hoever, the YJNPP built in high-level areas is one of the few examples in the world. On the premise of ensuring the development of nuclear power and economic development, it is impotant to understand the radioactive level and the potential radiation risk systematically and scientifically. Our resluts sugguested that the radioactive levels were too low to affect human health. There are differencs in the concentrations of radionuclides in different species of organisms, even in the same species from various habitats. The disribution of natual radionuclides in the organisms were associated with the living environments. The anthropogenic radionuclides still originated from global fallout. Our results are useful for understanding the effects of low dose radiation on human in the HBRAYJ at present. It is useful to provide some basic data and assess the radiological risk from the HBRAYJ and YJNNP in future. In particular, it is also essential for understanding the effects of low dose radiation on non-human species in future. 

In our paper, some other errors including the English language and style have been modifiedas follows:

  • Page 1/14,Line 18, “annual effective doseswas estimated in the high natural radioactive background areas” has been changed into “annual effective doses were estimated in the high natural radioactive background areas”.
  • Page 1/14,Line 19, “The specfic activities”has been changed into “The specific activities”.
  • Page 1/14,Line 21, “90Sr and gross beta special activities” has been changed into “ 90Sr and gross beta specific activities”.
  • Page 2/14,Line 48, “that can will ”has been changed into “that can ”.
  • Page 2/14,Line 51, in “at past.The YJNPP”, a space was added. In other place, a space was also added.
  • Page 2/14,Line 75, “from December 2012 to August 2012”has been changed into “from August 5th, 2011 to December 3rd, 2012 ”. 
  • Page 3/14,Line 92, in ”The ash were milled, sifted using an 80 mesh”, the word of “were”has been changed into “was”.
  • Page 5/14,Line 161, in the sentence of “1.2. Sweet potatoesform Dongping, Dagou and Huangjiangwei (Potatoes, n=3)”, the word “form” has been changed in to “ from”. In other places, the word “form” has been modified, e.g.,Page 5/14,Line 174,  Page 8/14,Line 201,  Page 7/14,Line 218.
  • Page 7/14,Line 218,  ”Freshwater fish (Tilapia)(n=1) fin Dongping...” has been changed in to  ”Freshwater fish (Tilapia) (n=1) in Dongping...”.
  • Page 7/14,Line 164, ”The high-low orders areas follows...” has been changed in to  ”The orders of the specific activities are as follows...”. In other places, the sentence has been modified, in Page 7/14,Line 176-177, Line 203, Page 8/14,Line 227, Page 8/14,Line 249,  Line 273.
  • Page 9/14,Line 292, “There were some differences between each radionuclides in various species” has been changed in to “There were some differences between each radionuclide in various species” .  
  • Page 10/14,Line 319,  ”...228Thwas...”has been changed in to ”...228Th was...”.
  • Page 12/14,Line 400, “On the premise of ensuring the development of nuclear power and economic development”has been changed in to “On the premise of ensuring the nuclear power and economic development”.
  • Page 12/14,Line 400, the word “impotant ”has been changed in to “important ”.
  • Page 12/14,Line 408 “Yangjiang nuclear power plants”has been changed in to “YJNNP”.

Reviewer 3 Report

Thank you for your prompt response to the peer review.   

Author Response

R: OK.

In our paper, some other errors including the English language and style have been modifiedas follows:

  • Page 1/14,Line 18, “annual effective doseswas estimated in the high natural radioactive background areas” has been changed into “annual effective doses were estimated in the high natural radioactive background areas”.
  • Page 1/14,Line 19, “The specfic activities”has been changed into “The specific activities”.
  • Page 1/14,Line 21, “90Sr and gross beta special activities” has been changed into “ 90Sr and gross beta specific activities”.
  • Page 1/14,Line 37-38, “where six nuclear power units (6×1080 MW) will been built in future”has been changed into “where six nuclear power units (6×1080 MW) have been built at present.”
  • Page 1/14,Line 41-42,“Five nuclear power units hadbeen put into commercial operation until May 25, ”has been changed into “At present, six nuclear power units had been put into commercial operation until July 24, 2019.”
  • Page 1/14,Line53-55,”...rare in the world. It is a necessary task to investigate systematically the current situation of radioactivity and assess scientifically the potentialrisk near the YJNPP near the HBRAY.” has been changed into  “...are in the world. The effects of low dose radiation on human in the HBRAYJ has been still an unsolved but very important issue. Also, the effects of low dose radiation on non-human species has become a concern since the beginning of this century. So it is a necessary task to investigate systematically the current radioactive levels and assess scientifically the potential risk.”
  • Page 2/14,Line 48, “that can will ”has been changed into “that can ”.
  • Page 2/14,Line 51, in “at past.The YJNPP”, a space was added. In other place, a space was also added.
  • Page 2/14,Line 59-70, ”Here,some organisms were collected according to the local production and diet habits in Yangjiang and the adjacent areas. This radioactive investigation in various species organisms is firstly and systematically conducted before and after the YJNPP commercial operation. Some items of 238U, 228Th, 226Ra, 40K, 90Sr, 137Cs, 134Cs, 51Cr, 55Fe, 54Mn, 58Co, 60Co, 65Zn, 110mAg and gross beta were analyzed to estimate annual effective dose near the HBRAYJ and YJNPP. Our dose calculations should help the public quantitatively assessing the risk associated with consumption of these popular organisms. Just our survey was conducted before and after the YJNNP operation, so it is expected that the results is useful to the routine and emergency radioactive monitoring around the YJNNP. This study is also expected to provide some basic data and help to assess the radiological risk from the HBRAYJ and Yangjiang nuclear power plants in ” has been changed in to

 “Here, some organisms were collected in Yangjiang and the adjacent areas during August 2011- April 2012, according to the local production and diet habits. This radioactive investigation in various species organisms is conducted firstly and systematically. Some items of 238U, 228Th, 226Ra, 40K, 90Sr, 137Cs, 134Cs, 51Cr, 55Fe, 54Mn, 58Co, 60Co, 65Zn, 110mAg and gross beta were analyzed to estimate annual effective dose near the HBRAYJ and YJNPP. Just sampling time is after the Fukushima Nuclear Power Plants Accident (FNA) in Marth 2011, and before the YJNPP commercial operation. So our results are expected to provide some basic background data. The dose calculations should help the public quantitatively assessing the risk associated with consumption of these popular organisms. It is useful to the routine and emergency radioactive monitoring around the YJNNP. This study is also expected and help to assess the radiological risk from the HBRAYJ and Yangjiang nuclear power plants in future.”    

  • Page 2/14,Line 75, “from December 2012 to August 2012”has been changed into “from August 5th, 2011 to December 3rd, 2012 ”. 
  • Page 3/14,Line 92, in ”The ash were milled, sifted using an 80 mesh”, the word of “were”has been changed into “was”.
  • Page 5/14,Line 161, in the sentence of “1.2. Sweet potatoesform Dongping, Dagou and Huangjiangwei (Potatoes, n=3)”, the word “form” has been changed in to “ from”. In other places, the word “form” has been modified, e.g.,Page 5/14,Line 174,  Page 8/14,Line 201,  Page 7/14,Line 218.
  • Page 7/14,Line 218,  ”Freshwater fish (Tilapia)(n=1) fin Dongping...” has been changed in to  ”Freshwater fish (Tilapia) (n=1) in Dongping...”.
  • Page 7/14,Line 164, ”The high-low orders areas follows...” has been changed in to  ”The orders of the specific activities are as follows...”. In other places, the sentence has been modified, in Page 7/14,Line 176-177, Line 203, Page 8/14,Line 227, Page 8/14,Line 249,  Line 273.
  • Page 9/14,Line 292, “There were some differences between each radionuclides in various species” has been changed in to “There were some differences between each radionuclide in various species” .  
  • Page 10/14,Line 319,  ”...228Thwas...”has been changed in to ”...228Th was...”.
  • Page 10/14,Line 343-344,another paragraph was added as follows:

Natual radionuclides of 238U, 228Th, 226Ra, and 40K in the organisms were originated from the surrounding soils. 228Th was significantly higher than 238U and 226Ra of natural U series in organisms due to the rich-Th soils in the HBRAYJ. For anthropogenic radionuclides, the YJNNP is not yet operational and could not have released anthropogenic pollutants. 137Cs and 90Sr should originated from global fallout. Althrough our survey was conducted after in Marth 2011, we think that The effect from the Fukushiama nuclear accident (FNA) on the organisms in our study also very limited and negliglible. 

  • Page 12/14,Line 400, “On the premise of ensuring the development of nuclear power and economic development”has been changed in to “On the premise of ensuring the nuclear power and economic development”.
  • Page 12/14,Line 400, the word “impotant ”has been changed in to “important ”.
  • Page 12/14,Line 408 “Yangjiang nuclear power plants”has been changed in to “YJNNP”.
  • The conclusionhas been modified without some discussions as follows:

Nuclear power is an effective way to achieve the goal of carbon dioxide emission reduction and realize the coordinated development of energy, economy and environment (3E system). At present, the YJNPP operation ensures energy needs and promotes sustainable economic development in Yangjiang and the adjacent areas. Hoever, the YJNPP built in high-level areas is one of the few examples in the world. On the premise of ensuring the development of nuclear power and economic development, it is impotant to understand the radioactive level and the potential radiation risk systematically and scientifically. Our resluts sugguested that the radioactive levels were too low to affect human health. There are differencs in the concentrations of radionuclides in different species of organisms, even in the same species from various habitats. The disribution of natual radionuclides in the organisms were associated with the living environments. The anthropogenic radionuclides still originated from global fallout. Our results are useful for understanding the effects of low dose radiation on human in the HBRAYJ at present. It is useful to provide some basic data and assess the radiological risk from the HBRAYJ and YJNNP in future. In particular, it is also essential for understanding the effects of low dose radiation on non-human species in future. 
